# WASSERSTEIN-GUIDED SYMBOLIC REGRESSION: MODEL DISCOVERY OF NETWORK DYNAMICS

## ABSTRACT

Real-world complex systems often miss high-fidelity physical descriptions and are typically subject to partial observability. Learning dynamics of such systems is a challenging and ubiquitous problem, encountered in diverse critical applications which require interpretability and qualitative guarantees. Our paper addresses this problem in the case of probability distribution flows governed by ODEs. Specifically, we devise a *white box* approach -dubbed Symbolic Distribution Flow Learner (`SDFL`)- combining symbolic search with a Wasserstein-based loss function, resulting in a robust model-recovery scheme which naturally lends itself to cope with partial observability. Additionally, we furnish the proposed framework with theoretical guarantees on the number of required *snapshots* to achieve a certain level of fidelity in the model-discovery. We illustrate the performance of the proposed scheme on the prototypical problem of Kuramoto networks and a standard benchmark of single-cell RNA sequence trajectory data. The numerical experiments demonstrate the competitive performance of `SDFL` in comparison to the state-of-the-art.

## 1 INTRODUCTION

Complex systems often emerge as large-scale networks of physical and societal interactions, with examples from epidemics to consensus dynamics, and from power grids to biological organisms. Despite their omnipresence, they often defy high-fidelity and fine-grain mathematical descriptions. The pursuit of accurate symbolic expressions that describe the evolution of such dynamical systems, is thus of paramount importance in many areas of science and engineering. Indeed, such parsimonious model descriptions offer several advantages, including compactness, explicit interpretations and high-fidelity generalization. Unlike *black-box* approaches, they give explicit insights on the underlying processes. Most notably, they allow for safety related qualitative guarantees such as asymptotic stability, which are crucial for critical applications.

Yet complex systems are prone to high-dimensionality and uncertainty, which render deriving explicit representations particularly challenging. This is often due to the difficulty of keeping track of non-linear relationships between a large number of variables, presence of noise, and scarcity of data. Hence, devising robust and efficient learning methods to uncover equations directly from data -also known as symbolic regression- is of great interest.

Symbolic regression has been extensively studied over recent years in both algebraic and differential equation discovery contexts. Common approaches include: sparse regression Brunton et al. (2016); Rudy et al. (2017); Chen et al. (2021); Kubalík et al. (2023), sequence-to-sequence deep neural network modelling Becker et al. (2022); Vastl et al. (2022); Biggio et al. (2021), and symbolic search-based formulations Cornforth & Lipson (2012); Gaucel et al. (2014); Cazenave (2013); Lu et al. (2021); Sun et al. (2023). One important setting, which has not been studied much though, is the discovery of dynamical systems governing probability distribution flows using only a few sampled screen shots, which is particularly

relevant for many applications ranging from epidemics modelling to cellular evolution prediction Bunne et al. (2022). Consequently, we propose to address this question, in the case of network flows, by designing a suitable equation recovery framework and illustrating its numerical performance on both synthetic and real-world data. This setup of symbolic learning of probability flows in networks entails several challenges:

- *Snapshots*, arriving at random times, have to be described in a continuous-time probabilistic representation.
- A robust inference algorithm enabling efficient search for an accurate solution -under partial observability- has to be designed.
- Permutation invariance, which is inherent in network structures, should be incorporated into the symbolic search, in particular to reduce the computational cost.

These challenges are addressed by our technical contributions:

- We combine the neural ODE framework Chen et al. (2018) with a symbolic search approach - respecting the permutation invariance- resulting in a *white-box* model for *trajectory inference*.
- We devise a suitable loss function that leverages the robustness properties of the Wasserstein distance Villani (2009), while taking into account the limited observability of the system under study.
- We back our devised algorithm with suitable sample complexity theoretical results.
- We demonstrate the performance of the proposed approach on prototypical problem of Kuramoto networks and a standard benchmark of single-cell population trajectory data.

The rest of the paper is organized as follows. In section 2, we review the related works and partly motivate our design choices. In section 3, we introduce the notions upon which the proposed algorithm is based. In section 4, we outline the main contributions of this work and provide the theoretical results that guided the design. In sections 5, we present numerical evaluations, followed by concluding remarks in section 6.

## 2 RELATED WORKS

**Symbolic search.** Symbolic regression was initially formulated as a discrete optimization problem Augusto & Barbosa (2000); Smits & Kotanchek (2005); Cornforth & Lipson (2012); Cazenave (2013), where the goal is to find the most accurate mathematical expression based on a predefined set of elementary operations and functions (e.g. $+, -, \times, \sin, \exp$ ). The mathematical expressions were represented through a one-to-one correspondence with *pre-order traversal* trees. For the resolution, Genetic Programming (GP) heuristics Koza (1994) were used to recover the equations underlying the training data. This method subsequently inspired applications in population evolution modelling Bongard & Lipson (2007), prediction of solar power production Quade et al. (2016), and Eulerian fluid flow hidden parameterization discovery Vaddireddy et al. (2020). Yet, GP suffers from a number of issues including over-fitting, brittleness to noise and poor-scalability Brunton et al. (2016); Petersen et al. (2019); Lu et al. (2021). A more recent approach Petersen et al. (2019) uses a Deep Reinforcement Learning (DRL) model to solve the optimization problem and generally outperforms GP-based models. However, deep learning models require large amounts of data in addition to their relative lack of full-automation, since architecture hyper-parameters have to be tuned by a human expert. On the other hand, a parallel well performing approach with the potential to achieve full-automation is Monte-Carlo Tree Search (MCTS). Based on the tree representation used in GP formulations, MCTS builds upon exploration-exploitation trade-off insights from sequential learning Munos et al. (2014). It is designed for a stochastic setting where data-points are costly to obtain and therefore naturally handles noisy input and scare-data settings well. Some works have applied MCTS successfully to simple Cazenave (2013); Islam et al. (2018); Lu et al. (2021) and broader Sun et al. (2023)

symbolic regression problems. The former used MCTS to uncover non-linear expressions in a supervised setting, whereas the latter also applied it to Ordinary Differential Equation (ODE) discovery when the initial conditions are deterministic. In the case of network structured problems, Shi et al. (2023); Cranmer et al. (2020) are the only works -to the best of our knowledge- which distill explicit equations. That is done by training a Graph Neural Network (GNN) and distilling equations for the message-passing operators. Additionally, only deterministic physical systems for which the whole trajectory is observed are considered. In contrast, we address a setting, where the goal is to identify the governing equations of *stochastic* network dynamics (with randomness in the initial condition) from the observation of a reduced number of screen shots across time.

**Sparse regularization.** One common property that emerges when modelling different natural phenomena and engineering problems is sparsity. Leveraging this fact, a number of works Brunton et al. (2016); Schaeffer (2017); Rudy et al. (2017); Loiseau & Brunton (2018); Rudy et al. (2019); Chen et al. (2021) formulate (deterministic) dynamic discovery as a $\ell_1$-regularized linear regression problem, over a predefined dictionary of basis functions. Subsequently, Boninsegna et al. (2018); Huang et al. (2022) extend this approach to the discovery of Stochastic Differential Equations (SDEs), using the Kramers-Moyal formula. From a theoretical perspective, Tran & Ward (2017) addresses the setting of highly corrupted data and provides conditions under which the underlying (polynomial) ODE can be recovered through $\ell_1$-regularized regression. On the other hand, Schaeffer et al. (2018) studies the question of minimal number of screenshots required to recover multivariate quadratic ODEs, in the case of random initial. However, they assume velocities to be known which limits their approach for the small and scattered data setting. Overall, although usefulness of sparsity-promoting approaches has been extensively demonstrated, they still rely on prior knowledge to define one of the main components, namely the function library. Whereas, if a library of massive size is chosen, the algorithm empirically fails to hold the sparsity constraint Sun et al. (2023). In comparison, we apply MCTS for our symbolic search part, which is not bound by such constraints.

**Sequence to sequence models.** The most recent approach to symbolic regression leverages the success of deep learning in sequential data modeling, such as in natural language processing Devlin et al. (2018). More precisely, Biggio et al. (2021); Kamienny et al. (2022); Vastl et al. (2022) propose a transformer-based architecture, which is trained to output mathematical expressions based on data-sets of feature-prediction pairs. That is, for each input data-set $\{(x_i, y_i)\}_{i=1}^n$, the model outputs an expression $e$ corresponding to a function $f_e$ satisfying $\forall i \in [1, n], \ y_i \simeq f_e(x_i)$. One challenge though, is to generate a training data-set $\left\{ (\{(x_i^j, f_{e_j}(x_i^j))\}_{i=1}^n, f_{e_j}) \right\}_{j=1}^N$ which is rich enough to represent parsimonious equations that are frequently encountered in practice. To achieve that, equations are generated as binary trees, following the work of Lample & Charton (2020); Kusner et al. (2017) and previously mentioned symbolic search approach representations, e.g. Cazenave (2013). On the other hand, Li et al. (2019) combine a recurrent neural network with MCTS to enforce asymptotic constraints on the learned expression, while some works Martius & Lampert (2016); Sahoo et al. (2018); Costa et al. (2020); Kubalík et al. (2023) propose to use different elementary mathematical operators (e.g. $+, \times, \cos\ldots$ etc) as activation functions for neural net architectures, while imposing sparsity on the parameters to extract interpretable analytic expressions. However, these approaches rely on the generation of large data-sets, which is prohibitively costly for high-dimensional ODE/PDEs.

## 3 PROBLEM FORMULATION AND BACKGROUND

### 3.1 GENERAL SETUP

Fix $T > 0$ as a positive time horizon, and consider the random variable $\mathbf{x}_0 \in \mathbb{R}^d$ to be the initial condition of the state variable $\mathbf{x}_t$ which evolves according to the ODE system

$$\dot{\boldsymbol{y}} = \boldsymbol{f}(\boldsymbol{y}) \tag{1}$$

in time $t \in [0, T]$. In our setting, $\boldsymbol{f}$ encodes the interactions between different components of $\mathbf{x}_t$ over time, which are distributed according a known and fixed network topology $\mathcal{G}$. We denote by $\mu_t$, the probability measure induced by $\mathbf{x}_t$ i.e. $\mathbf{x}_t \sim \mu_t$, and by $p_t$ the corresponding probability density. The latter evolves -if $\mu_t$ is absolutely continuous with respect to the Lebesgue measure- according to the equation

$$\log p_t(\mathbf{x}_t) = \log p_0(\mathbf{x}_0) - \int_0^t \mathrm{Tr}\left(\frac{\mathrm{d}\boldsymbol{f}}{\mathrm{d}\boldsymbol{y}}(\boldsymbol{x}_s)\right) \, ds \tag{2}$$

along the trajectory $(\mathbf{x}_s)_{s \in [0,T]}$ (see e.g. Chen et al. (2018)). For $n \geq 2$ measurement times $\{t_0, t_1, \ldots, t_n\}$, assume $m \geq 1$ samples of each of the distributions $\mu_{t_0}, \mu_{t_1}, \ldots, \mu_{t_n}$ are observed, resulting in *snapshots* represented by the corresponding empirical probability measures $\{\hat{\mu}_{t_1,m}, ..., \hat{\mu}_{t_n,m}\}$. The question addressed in this paper is that of the recovery of the function $\boldsymbol{f}$ defining the ODE that governs the given dynamics. That is, *trajectory inference* Hashimoto et al. (2016); Tong et al. (2020); Bunne et al. (2022); Huguet et al. (2022), through an explicit closed form ODE. For that matter, the solution of the ODE can be seen as the image of $\boldsymbol{f}$ by the operator given by

$$\boldsymbol{F}(\boldsymbol{f}) : (t, \boldsymbol{x}_0) \mapsto \boldsymbol{F}^t(\boldsymbol{f})(\boldsymbol{x}_0)$$

where $t \mapsto \boldsymbol{x}_t = \boldsymbol{F}^t(\boldsymbol{f})(\boldsymbol{x}_0)$ is the solution of the Cauchy problem:

$$\begin{cases} \dot{\boldsymbol{y}} = \boldsymbol{f}(\boldsymbol{y}) \\ \boldsymbol{y}(0) = \boldsymbol{x}_0 \end{cases} \tag{3}$$

for a given initial condition $\boldsymbol{x}_0 \in \mathbb{R}^d$. Note that, the problem of recovering $\boldsymbol{f}$ from $(\hat{\mu}_{t_i,m})_{i \in \{1,...,n\}}$ is *ill-posed* in general, and further assumptions are needed. Such assumptions should represent prior information that can act as a form of regularization rendering the learning task at hand more feasible. In our case, the required prior information comes from the fact that the estimator of $\boldsymbol{f}$ is constructed by combining analytic expressions from a fixed pre-selected set. We make the assumption that the observation instants fulfill $\forall i \in \{0, \ldots, n\}$, $t_i \in (iT/n, (i+1)T/n)$. A straight-forward extension, though, can be obtained for uniformly sampled time instants $(t_i)_{i \in \{0,...,n\}}$ based on a Quasi-Monte Carlo scheme convergence argument Niederreiter (1978). To translate this setting to an optimization framework, first, we review some concepts and facts about distances in probability spaces upon which a suitable goodness-of-fit measure is proposed.

### 3.2 WASSERSTEIN GUIDANCE

Considering probability distribution flow modelling, we will need to compare predicted distributions with observed ones. Popular measures of disparity between probability distributions include the Kullback–Leibler divergence in the computational context, the total variation distance in the theoretical one, and the Wasserstein distance Villani (2009) in both. Indeed, due to its singular theoretical properties, e.g. weak topology metrization, convexity and robustness (see e.g. Mohajerin Esfahani & Kuhn (2018) for the latter), but also its amenability to efficient computation, the 2-Wasserstein distance $\mathcal{W}_2$ represents a natural choice for the design of a robust trajectory inference algorithm. Consider two measure spaces $(\mathcal{X}, \mu)$ and $(\mathcal{Y}, \nu)$ and denote by $\Pi(\mu, \nu)$ the set of their couplings, then $\mathcal{W}_2$ reads

$$\mathcal{W}_2(\mu, \nu) = \min_{\pi \in \Pi(\mu,\nu)} \int_{\mathcal{X} \times \mathcal{Y}} \|\alpha - \beta\|_2^2 \, \mathrm{d}\pi(\alpha, \beta) , \tag{4}$$

where $\| \, . \, \|_2$ is the Euclidean norm. If we had access to $(\mu_t)_{t\in[0,T]}$, the goodness-of-fit of a candidate estimator $\hat{\boldsymbol{f}}$ could be suitably defined as:

$$L(\hat{\boldsymbol{f}}) = \int_0^T \mathcal{W}_2(\boldsymbol{F}^t(\hat{\boldsymbol{f}})_{\#}\mu_0, \, \mu_t) \, \mathrm{d}t \,, \tag{5}$$

where $\boldsymbol{F}^t(\hat{\boldsymbol{f}})_{\#}\mu_0$ is the push-forward of the initial probability measure by the partial flow map $\boldsymbol{x}_0 \mapsto F^t(\hat{f})(\boldsymbol{x}_0)$. In other words, $L(\hat{\boldsymbol{f}})$ is the time aggregated Wasserstein distance between the measure resulting from the inferred trajectory and the one resulting from the ground-truth dynamics. However, since $\mu_t$ is only partially known, we work with an approximation of $L(\hat{\boldsymbol{f}})$ based on the observed *snapshots*, as discussed in the following, after presenting the basic building blocks of MCTS.

### 3.3 MONTE-CARLO TREE SEARCH FOR SYMBOLIC SEARCH

Due to the combinatorial nature of the state of possible mathematical expressions, it is crucial to reduce the number of required evaluations. To cope with that, MCTS targeted to symbolic search, relies on tree representations of mathematical expressions, where the nodes express unitary $(\sin(\cdot), \cos(\cdot), \dots)$ or binary operations $(+, -, \times, \dots)$ and leafs express variables and constants $(x_1, x_2, c, \dots)$. This is formalized using the notion of context-free grammars Kusner et al. (2017); Sun et al. (2023). Given such representation, the intuitive idea behind MCTS is to explore as few operators as possible in building the tree, yet to identify the optimum with a high probability. To achieve that, it leverages insights from Upper-Confidence Bound (UCB) algorithms in sequential learning Lattimore & Szepesvári (2020), where the goal translates to optimizing a function which measures both accuracy and under-exploration of a candidate operation. We refer to Sun et al. (2023) for a detailed pseudo-code, and recall the main components translated to our context below:

- **Stochastic roll-outs.** To complete a partially built tree i.e. a partial expression denoted by $\boldsymbol{s}$, the value of each potential additional node $a$ (representing an elementary operation) needs to be obtained, in order to choose the one with the highest value. For that matter, for each choice of node, perform random roll-outs of potential completions. Each roll-out will either lead to a complete expression $\hat{\boldsymbol{f}}(\boldsymbol{x})$, whose corresponding score function can be computed; or to an expression whose length exceeds a predefined maximal length $M$ leading to a score of $0$. The largest of the obtained scores is then stored in $V(s, a)$ to be used in the next step.

- **Exploration-aware selection.** Once all operations have been visited at least once, select the next one by maximizing a performance measure that takes into account under-exploration, and which can be taken in practice[1] as

$$UCT(\boldsymbol{s}, a) := V(\boldsymbol{s}, a) + c\sqrt{\ln[N(\boldsymbol{s})]/N(\boldsymbol{s}, a)}, \; c > 0$$

  where $V(\boldsymbol{s}, a)$ is the average value of operation $a$, given the operations constituting the current tree i.e. $\boldsymbol{s} = [a_0, a_1, \dots, a_{p-1}]$. Furthermore, $N(\boldsymbol{s}, a)$ is the number of times that operation $a$ was chosen at tree state $\boldsymbol{s}$ with $N(\boldsymbol{s}) = \sum_{a\in\mathcal{A}(\boldsymbol{s})} N(\boldsymbol{s}, a)$, where $\mathcal{A}(\boldsymbol{s})$ is the set of available operations at tree state $\boldsymbol{s}$. Note that the square root term $\sqrt{\ln[N(\boldsymbol{s})]/N(\boldsymbol{s}, a)}$ quantifies the under-exploration of operation $a$ at state $\boldsymbol{s}$.

## 4 TECHNICAL APPROACH

---

[1]For the sample complexity result presented in prop. 1 to hold though, the scheme should be based on the more involved expression for UCT described in Shah et al. (2020).

To minimize the introduced loss function, given by equation 5, we replace the neural net estimator in the neural ODE framework of Chen et al. (2018); Kidger (2022) with a search for an explicit analytic formula for $\boldsymbol{f}$. This is achieved through a symbolic regression algorithm instantiated by MCTS Sun et al. (2023). The devised workflow -of inferring the network dynamics from given snapshots- is illustrated in Fig. 1. From a guarantee perspective, in addition to a bound on the approximation error of $L(\hat{f})$ by its empirical estimate $\hat{L}_{m,n}(\hat{f})$ for a given $\hat{f}$, we quatify the sample complexity of MCTS, using regret bound-based results from bandit theory Lattimore & Szepesvári (2020).

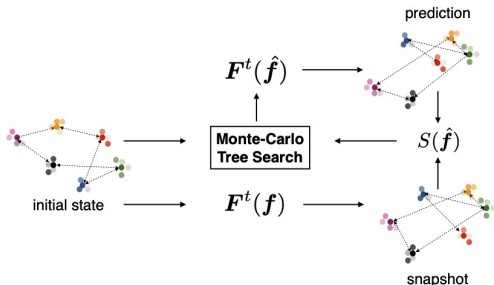

Figure 1: The devised model-discovery setup

### 4.1 DISCRETE LOSS FUNCTION

The convergence of the discrete version of the loss function, defined for continuously differentiable $\hat{\boldsymbol{f}} : \mathbb{R}^d \to \mathbb{R}^d$ by

$$\hat{L}_{m,n}(\hat{\boldsymbol{f}}) = \frac{1}{n} \sum_{i=1}^{n} \mathcal{W}_2(\boldsymbol{F}^{t_i}(\hat{\boldsymbol{f}})_{\#}\hat{\mu}_{t_0,m}, \hat{\mu}_{t_i,m}),$$

to the continuous one i.e. $L$, is controlled by the number of snapshots and the size of the snapshot sample set. We build upon results of Fournier & Guillin (2015); Bonnans & Shapiro (2013) about *finite-sample* rates of convergence of the empirical measure in the Wasserstein space, and regularity of the Wasserstein distance to obtain the following theorem, where the proof is postponed to Appendix A.

**Theorem 1.**
Let $(\mu_t)_{t \geq 0}$ have a compact support and $\hat{f} : \mathbb{R}^d \longrightarrow \mathbb{R}^d$ be a differentiable function. Assume that $t \mapsto \mu_t$ is differentiable. Then, for all $m, n \geq 2$,

$$\mathbb{E}\left| \frac{1}{n} \sum_{i=1}^{n} \mathcal{W}_2(\boldsymbol{F}^{t_i}(\hat{\boldsymbol{f}})_{\#}\hat{\mu}_{t_0,m}, \hat{\mu}_{t_i,m}) - \frac{1}{T} \int_0^T \mathcal{W}_2(\boldsymbol{F}^t(\hat{\boldsymbol{f}})_{\#}\mu_0, \mu_t)\, \mathrm{d}t \right| = O\left( \frac{1}{m^{1/2}} + \frac{1}{n} \right),$$

where $\hat{\mu}_{t_i,m}$, for $i \in \{1, \ldots, n\}$, denotes the empirical measure corresponding to the $m$ observed realizations of each of the probability distributions $(\mu_{t_i})_{1 \leq i \leq n}$.

### 4.2 SYMBOLIC FLOW DISCOVERY ALGORITHM

Given the discrete loss $\hat{L}_{m,n}$, the model-recovery algorithm is based on computing predicted snapshots corresponding to a candidate estimate $\hat{\boldsymbol{f}}$ and comparing them to the observed ones. The predictions will be determined by solving the obtained ODE, i.e. equation 1 where $\boldsymbol{f}$ is replaced by $\hat{\boldsymbol{f}}$, through a numerical integration scheme, such as Runge–Kutta solvers. In parallel, the optimiza-

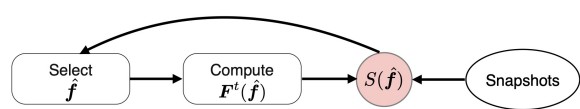

Figure 2: Building blocks of the proposed algorithm

tion over candidate estimates is realized using a MCTS-based symbolic search. For that matter, the goal of MCTS will be to maximize a score function defined for a differentiable $\hat{\boldsymbol{f}} : \mathbb{R}^d \longrightarrow \mathbb{R}^d$ by

$S(\hat{f}) = 1/(1 + \hat{L}_{m,n}(\hat{f}))$. To ensure permutation invariance in the obtained expressions, which is characteristic of network systems, we restrict the search space by applying a permutation invariant aggregation operation (e.g. *sum*) to each expression before each evaluation. We report an ablation study corresponding to the removal of the permutation invariance module in Appendix D.2. The pseudo-code of the algorithm is summarized in *Algorithm* 1, where the main steps are illustrated in Fig. 2, with more details in Appendix E.

---

**Algorithm 1** Symbolic Distribution Flow Learner

---

**Inputs:** Number of episodes $N$, number of roll-outs $H$, screen-shots $(\hat{\mu}_{t_i,m})_{i,m}$ at $(t_i)_{0 \leq i \leq n}$
**Initialization:** Estimate the value of each operation $(+, -, \times, \sin, \dots)$ as a root node ;
**for** $e = 1, \dots, N$ **do**:
    Randomly select a root node and build an expression tree as follows:
    **if** Tree is complete **then**
        Evaluate the corresponding estimate $\hat{f}$ by computing $S(\hat{f})$, Back-propagate
        the obtained value ;
    **else**
        Run $H$ roll-outs, Store the best estimate, Back-propagate the corresponding value ;
        Select the operation $a$ maximizing $UCT(s, a)$ where $s$ is the current state of the tree ;
    **end if**
**end for**
**Return:** Most accurate $\hat{f}$ over the $N$ episodes ;

---

### 4.3 MONTE-CARLO TREE SEARCH SAMPLE COMPLEXITY

A crucial question in MCTS-based algorithms is to estimate the number of required episodes, executed by the algorithm, in order to honor a certain error tolerance in the obtained solution of the target optimization problem. We apply a non-asymptotic error analysis result by Shah et al. (2020) to determine the minimal number of episodes -also known as sample complexity- that ought to be used (see Appendix B for the proof).

**Proposition 1.**
The number of score evaluations $E$ required for the MCTS algorithm[2] to find an $\varepsilon$-optimal[3] solution where $\varepsilon > 0$, is at most given by:

$$E = O\left(q \cdot \varepsilon^{-(4+M)} \cdot (\log \frac{1}{\varepsilon})^5\right) ,$$

where $M$ is the maximum allowed expression length and $q$ the size of the chosen elementary function set.

## 5 NUMERICAL EXPERIMENTS

First, we illustrate the performance of the proposed algorithm on the Kuramoto network system of ODEs, used across the biological, chemical and electrical domains to model circadian oscillators, pacemaker cells in the heart and electrical power networks among other applications Discacciati & Hesthaven (2021); Dörfler & Bullo (2014). Then, we conduct an evaluation on a real-world dataset of embryoid stem cell trajectories Moon et al. (2019). We provide comparisons of our algorithm with the current trajectory inference state-of-the-art algorithms, namely `TrajectoryNet` Tong et al. (2020) and `JKOnet` Bunne et al. (2022).

---

[2] With UCT defined as in Shah et al. (2020).
[3] An $\varepsilon$-optimal solution of an optimzation problem $\min_{x \in E} g(x)$ is a value $x_\varepsilon \in E$ that satisfies $g(x_\varepsilon) \leq \min_{x \in E} g(x) + \varepsilon$.

`TrajectoryNet` relies on a (neural net-based) continuous normalizing flow Grathwohl et al. (2018) formulation, augmented with relevant regularizations such as growth rate and velocity penalization. `JKOnet` builds upon the celebrated JKO scheme Jordan et al. (1998) describing energy gradient flows, where it parameterizes the energy function and the Monge potential using Input Convex Neural Networks Amos et al. (2017). In addition to being based on black-box models -in this instance, neural nets- these approaches contrast with `SDFL` in that they require (extra)-hyper-parameter tuning. For the experimental comparison, we retrain the models with the architectures and hyper-parameters proposed by the respective authors Tong et al. (2020); Bunne et al. (2022); however, we employ early-stopping to avoid over-fitting to the smaller data-sets. For `JKOnet`, we use a small regularization parameter $\varepsilon = 0.001$ to make its target closer to the Wasserstein distance. We conduct experiments in the small and noisy data regime with varying sample sizes (per snapshot), illustrating the suitability of `SDFL` for practical costly data-collection conditions. Similar to previous studies, numerical evaluations are conducted based on the Wasserstein distance between the predicted empirical distributions and the ground-truth distributions (averaged over 3 runs). The equations obtained by `SDFL` are reported in Appendix C, and the computational time in Appendix D.1.

## 5.1 Kuramoto system of ODEs

We investigate the recovery of the Kuramoto system with the state variable $\boldsymbol{x}_t = (\theta_i(t))_{1 \leq i \leq d}$ following

$$\dot{\theta}_i(t) = \omega_i + \frac{1}{d} \sum_{j=1}^{d} K_{ij} \sin(\theta_j(t) - \theta_i(t)),$$

where $(\omega_i)_{1 \leq i \leq d}$ are the corresponding natural frequencies. Gaussian initial condition (with mean 2 and unity variance) is employed. Moreover $\mathcal{G} = (K_{ij})_{1 \leq i, j \leq d}$ is the graph weight matrix. For simplicity, following Discacciati & Hesthaven (2021), we assume a fully connected uniformly weighted graph (i.e. we take $K_{ij} = K$ for all $i, j \in \{1, \ldots, d\}$). We consider $n = 15$ snapshots, in $d = 3$ dimensions, over the time horizon $T = 30$. Furthermore the natural frequencies are set to 0.01. To highlight the accuracy and robustness of our approach, Figure 3 illustrates the inferred trajectory of state variable (using `SDFL`), and robustness of Wasserstein distance with respect to the noise. Note that to reduce the `SDFL` running time, it

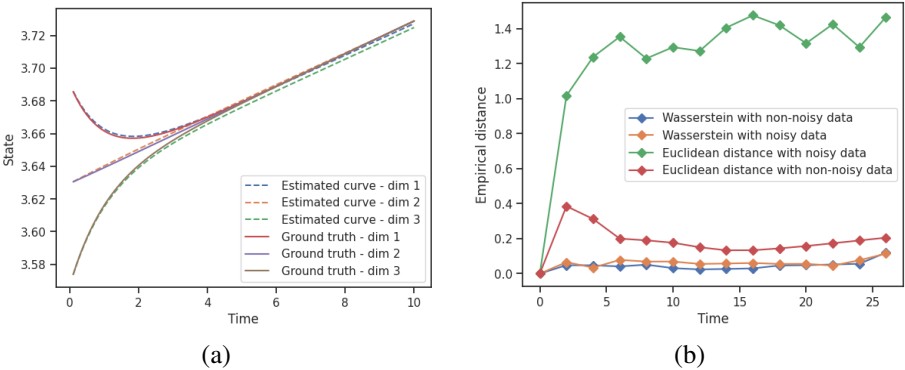

Figure 3: (a) State trajectory (b) Distance between inferred and reference distributions, by different metrics.

was run in two steps where the global function structure is fitted first, followed by the estimation of regression coefficients (see Appendix C for details). The results reported in table 1 suggest `SDFL` is competitive with the state-of-the-art, while the performance for the different methods improves with increasing sample size.

| Training sample size | SDFL | JKONet | TrajectoryNet |
|---|---|---|---|
| 50 | $0.5733 \pm 0.04$ | $0.8688 \pm 0.22$ | $4.3315 \pm 0.13$ |
| 100 | $0.5733 \pm 0.06$ | $0.7699 \pm 0.18$ | $4.8377 \pm 0.10$ |
| 150 | $0.4612 \pm 0.05$ | $0.7621 \pm 0.22$ | $3.1540 \pm 0.11$ |

Table 1: Prediction error comparison in the Wasserstein metric for the Kuramoto model

## 5.2 SINGLE-CELL POPULATION DYNAMICS

We consider the problem of learning the evolution of embryonic stem cells based on single-cell RNA sequencing data measurements over a period of 27 days, where the data was collected at 5 different snapshots (see Moon et al. (2019)). Apart from the high cost of obtaining a large dataset, a key difficulty is that a cell is (usually) destroyed during a measurement Bunne et al. (2022). Hence, there is a need for schemes which predict the distribution evolution across time, rather than individual trajectories, from limited observations. The dataset was pre-processed using the recent dimensionality reduction technique PHATE Moon et al. (2019), which is specifically designed to preserve maximum variablity in the low-dimensional space, while allowing for intuitive visualization, and therefore interpretation. Similar to the previous case of Kuramoto network, SDFL outperforms the benchmarks. This is encouraging, specifically since for the cell population dynamics, the dataset is not produced based on an underlying mathematical model. Most importantly, it confirms the relevance of SDFL as a dynamic regression tool for real-world data subject to noise and limited observability. It should be emphasized that neural net based approaches (TrajectoryNet and JKONet) require sufficiently large amounts of data to reach their optimal performance. As for JKONet, one explanation for why it performs better than TrajectoryNet, is that it enforces an inductive bias through the JKO scheme Bunne et al. (2022). The performance of the SDFL model is stable across training sample sizes, as we illustrate its competitiveness even when no regression parameters are fitted (see appendix C).

| Training sample size | SDFL | JKONet | TrajectoryNet |
|---|---|---|---|
| 100 | $2.0822 \pm 0.31$ | $3.1847 \pm 0.54$ | $6.7841 \pm 0.26$ |
| 200 | $2.0822 \pm 0.31$ | $3.7277 \pm 0.51$ | $6.6531 \pm 0.06$ |
| 300 | $2.0822 \pm 0.31$ | $3.0355 \pm 0.70$ | $6.2591 \pm 0.11$ |

Table 2: Prediction error comparison in the Wasserstein metric for embryoid stem cell trajectory

## 6 CONCLUSION

We investigated model-discovery of dynamic systems, subject to randomness. In particular, leveraging a symbolic regression approach, we considered a setup in which only a few random observations of the system are provided in time and space. We proposed SDFL, incorporating several innovative ideas, in order to tackle this multifaceted dynamic inference problem. Specifically, an appropriate measure for goodness-of-fit was introduced by devising a time integrated Wasserstein loss. This design choice turned out to be powerful by offering robustness of the model-discovery in presence of noise/uncertainty in the data. In addition, we derived theoretical results on the model performance, along with guarantees on its sample complexity. Motivated by the performance of SDFL in our numerical experiments, we believe these developments contribute to paving the way for progress on robust model-discovery from noisy and limited data. The computational scaling of SDFL to the higher dimensions remain an interesting open problem for future studies.

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

# A PROOF OF THEOREM 1

*Proof.* Recall that we would like to prove there are positive constants $M_1, M_2 > 0$ such that:

$$\mathbb{E}\left|\frac{1}{n}\sum_{i=1}^{n}\mathcal{W}_2(\boldsymbol{F}^{t_i}(\hat{\boldsymbol{f}})_{\#}\hat{\mu}_{t_0,m}, \hat{\mu}_{t_i,m}) - \frac{1}{T}\int_0^T \mathcal{W}_2(\boldsymbol{F}^t(\hat{\boldsymbol{f}})_{\#}\mu_0, \mu_t)\,\mathrm{d}t\right| \leq \frac{M_1}{m^{1/2}} + \frac{M_2}{n}$$

provided $m, n \geq 2$. Without loss of generality, we assume $T = 1$. By proposition 5 and 7 in Fournier & Guillin (2015), given that $\boldsymbol{x}_0 \mapsto \boldsymbol{F}^t(\boldsymbol{x}_0)$ is continuous and for all $i \in \{1, \ldots, n\}$, $\mu_{t_i}$ has compact support on the Polish space $\mathbb{R}^d$, there exists $C_1, C_2 > 0$ such that

$$\mathbb{E}\left[\frac{1}{n}\sum_{i=1}^{n}\mathcal{W}_2(\boldsymbol{F}^{t_i}(\hat{\boldsymbol{f}})_{\#}\hat{\mu}_{t_0,m}, \boldsymbol{F}^{t_i}(\hat{\boldsymbol{f}})_{\#}\mu_{t_0})\right] \leq \frac{C_1}{m^{1/2}}$$

$$\text{and} \qquad \mathbb{E}\left[\frac{1}{n}\sum_{i=1}^{n}\mathcal{W}_2(\hat{\mu}_{t_i,m}, \mu_{t_i})\right] \leq \frac{C_2}{m^{1/2}}.$$

Therefore, thanks to the triangular inequality satisfied by $\mathcal{W}_2$, we get

$$\mathbb{E}\left|\frac{1}{n}\sum_{i=1}^{n}\mathcal{W}_2(\boldsymbol{F}^{t_i}(\hat{\boldsymbol{f}})_{\#}\hat{\mu}_{t_0,m}, \hat{\mu}_{t_i,m}) - \frac{1}{n}\sum_{i=1}^{n}\mathcal{W}_2(\boldsymbol{F}^{t_i}(\hat{\boldsymbol{f}})_{\#}\mu_{t_0}, \mu_{t_i})\right| \leq$$

$$\mathbb{E}\left[\frac{1}{n}\sum_{i=1}^{n}\mathcal{W}_2(\boldsymbol{F}^{t_i}(\hat{\boldsymbol{f}})_{\#}\hat{\mu}_{t_0,m}, \boldsymbol{F}^{t_i}(\hat{\boldsymbol{f}})_{\#}\mu_{t_0})\right] + \mathbb{E}\left[\frac{1}{n}\sum_{i=1}^{n}\mathcal{W}_2(\hat{\mu}_{t_i,m}, \mu_{t_i})\right] \leq \frac{C_1 + C_2}{m^{1/2}}. \quad (6)$$

On the other hand, denoting by $\mathcal{M}(\mathbb{R}^d \times \mathbb{R}^d)$ the Banach space of finite measures on $\mathbb{R}^d \times \mathbb{R}^d$ equipped with the total variation norm, the mapping $(t, \gamma) \in [0, 1] \times \mathcal{M}(\mathbb{R}^d \times \mathbb{R}^d) \mapsto \int_{\mathbb{R}^d \times \mathbb{R}^d} \|\boldsymbol{F}^t(\hat{\boldsymbol{f}})(\boldsymbol{x}) - \boldsymbol{y}\|_2^2\,\mathrm{d}\gamma(\boldsymbol{x}, \boldsymbol{y})$ is differentiable. Moreover, by compactness of the set of couplings given two fixed marginals and by theorem 5.20 in Villani (2009), conditions $(ii)$ and $(iii)$ of theorem 4.24 in Bonnans & Shapiro (2013) are satisfied leading to the fact that the mapping $t \mapsto \mathcal{W}_2(\boldsymbol{F}^t(\hat{\boldsymbol{f}})_{\#}\mu_0, \mu_t)$ is continuously differentiable on $[0, 1]$[4]. Hence, there exists a positive constant $C_3 > 0$ such that

$$\left|\int_{(i-1)/n}^{i/n} \mathcal{W}_2(\boldsymbol{F}^t(\hat{\boldsymbol{f}})_{\#}\mu_0, \mu_t)\,\mathrm{d}t - \frac{1}{n}\mathcal{W}_2(\boldsymbol{F}^{t_i}(\hat{\boldsymbol{f}})_{\#}\mu_{t_0}, \mu_{t_i})\right| \leq \frac{C_3}{n^2}$$

for $i \in \{1, ..., n\}$, which yields

$$\left|\int_0^1 \mathcal{W}_2(\boldsymbol{F}^t(\hat{\boldsymbol{f}})_{\#}\mu_0, \mu_t)\,\mathrm{d}t - \frac{1}{n}\sum_{i=1}^{n}\mathcal{W}_2(\boldsymbol{F}^{t_i}(\hat{\boldsymbol{f}})_{\#}\mu_{t_0}, \mu_{t_i})\right| \leq \frac{C_3}{n}.$$

Consequently, by summation with (6), we get

$$\mathbb{E}\left|\frac{1}{n}\sum_{i=1}^{n}\mathcal{W}_2(\boldsymbol{F}^{t_i}(\hat{\boldsymbol{f}})_{\#}\hat{\mu}_{t_0,m}, \hat{\mu}_{t_i,m}) - \int_0^1 \mathcal{W}_2(\boldsymbol{F}^t(\hat{\boldsymbol{f}})_{\#}\mu_0, \mu_t)\,\mathrm{d}t\right| \leq \frac{C_1 + C_2}{m^{1/2}} + \frac{C_3}{n}.$$

$\square$

---

[4]Given that the optimal coupling is unique and that the directional derivative is continuous with respect to $t \in [0, 1]$ -as a supremum of a differentiable parametric family of convex function.

## B   PROOF OF PROPOSITION 1

*Proof.* First, denote by $(\mu_t)_{t \in [0,T]}$ the ground-truth probability flow and let $\boldsymbol{f}$, $\boldsymbol{g}$ be the continuously differentiable functions defined by two analytic expressions from the selection space of MCTS. Recall that the goal is to to maximize the functional given by

$$S : \boldsymbol{f} \mapsto \frac{1}{1 + L(\boldsymbol{f})} \ .$$

The result is obtained by showing that the conditions of theorem 2 in Shah et al. (2020) are satisfied in our setting. For that matter, it suffices[5] to show that

$$L : \boldsymbol{f} \mapsto \int_0^T \mathcal{W}_2(\boldsymbol{F}^t(\boldsymbol{f})_{\#}\mu_0, \ \mu_t) \ \mathrm{d}t$$

is Lipschitz with respect to the $L^1$ or $\| \cdot \|_\infty$ norms; since the derivative of $\boldsymbol{h} : \boldsymbol{x} \mapsto \frac{1}{1+\boldsymbol{x}}$ is bounded on $\mathbb{R}^+$ implying that $\boldsymbol{h}$ is Lipschitz. Furthermore, we have

$$|S(\boldsymbol{f}) - S(\boldsymbol{g})| \leq \int_0^T \left| \mathcal{W}_2(\boldsymbol{F}^t(\boldsymbol{f})_{\#}\mu_0, \ \mu_t) - \mathcal{W}_2(\boldsymbol{F}^t(\boldsymbol{g})_{\#}\mu_0, \ \mu_t) \right| \ \mathrm{d}t$$

$$\leq \int_0^T \mathcal{W}_2(\boldsymbol{F}^t(\boldsymbol{f})_{\#}\mu_0, \boldsymbol{F}^t(\boldsymbol{g})_{\#}\mu_0) \ \mathrm{d}t$$

because $\mathcal{W}_2$ is a distance. Additionally, since both distributions have compact support, and by regularity of $\boldsymbol{F}^t(\boldsymbol{f})$, $\boldsymbol{F}^t(\boldsymbol{g})$, there exists a constant $C > 0$ such that for all $t \in [0,T]$,

$$\mathcal{W}_2(\boldsymbol{F}^t(\boldsymbol{f})_{\#}\mu_0, \boldsymbol{F}^t(\boldsymbol{g})_{\#}\mu_0) \leq C \cdot \mathcal{W}_1(\boldsymbol{F}^t(\boldsymbol{f})_{\#}\mu_0, \boldsymbol{F}^t(\boldsymbol{g})_{\#}\mu_0)$$

$$\leq C \cdot \sup_{\mathrm{Lip}(q) \leq 1} \left| \int q \ \mathrm{d}(\boldsymbol{F}^t(\boldsymbol{f})_{\#}\mu_0) - \int q \ \mathrm{d}(\boldsymbol{F}^t(\boldsymbol{g})_{\#}\mu_0) \right|$$

$$\leq C \cdot \sup_{\mathrm{Lip}(q) \leq 1} \left| \int q \circ \boldsymbol{F}^t(\boldsymbol{f}) \ \mathrm{d}\mu_0 - \int q \circ \boldsymbol{F}^t(\boldsymbol{g}) \ \mathrm{d}\mu_0 \right|$$

$$\leq C \cdot \sup_{\mathrm{Lip}(q) \leq 1} \int \left| q \circ \boldsymbol{F}^t(\boldsymbol{f}) - q \circ \boldsymbol{F}^t(\boldsymbol{g}) \right| \ \mathrm{d}\mu_0$$

$$\leq C \cdot \int \left| \boldsymbol{F}^t(\boldsymbol{f}) - \boldsymbol{F}^t(\boldsymbol{g}) \right| \ \mathrm{d}\mu_0$$

$$\leq T \cdot C \cdot \|\boldsymbol{f} - \boldsymbol{g}\|_\infty$$

where the second inequality is justified by the dual representation of $\mathcal{W}_1$, the third by the change of variable formula[6], and the fifth by the Lipschitz property of $q$. The last inequality is justified by the fact that the solutions $t \mapsto \boldsymbol{F}^t(\boldsymbol{f})(\boldsymbol{x})$ and $t \mapsto \boldsymbol{F}^t(\boldsymbol{g})(\boldsymbol{x})$ of the differential equations $\dot{\boldsymbol{y}} = \boldsymbol{f}(\boldsymbol{y})$ and $\dot{\boldsymbol{y}} = \boldsymbol{g}(\boldsymbol{y})$ respectively, for a given initial condition $\boldsymbol{x} \in \mathbb{R}^d$, are fixed points of the Picard operator[7], which is Lipschitz.

Consequently, integrating over $t \in [0,T]$ on both sides, we obtain the Lipschitz property of $S$. Finally, since $L(\boldsymbol{f})$ can be approximated by $\hat{L}_{m,n}(\boldsymbol{f})$ with arbitrary accuracy, we get the sample complexity upper-bound. □

---

[5]Note that, since the considered state space is finite, we do not need to construct an explicit covering.

[6]Note that $q \circ \boldsymbol{F}^t(\boldsymbol{f})$ denotes the composition of $q$ by $\boldsymbol{F}^t(\boldsymbol{f})$.

[7]The Picard operator is introduced in the proof of the existence of (local) solutions to ODEs (see e.g. Coddington et al. (1956)).

## C  Implementation details & Reproducibility

For the implementation of SDFL, we set the building operations consisting of $\{+, -, \times, \div, \cos, \sin, \exp\}$ with a maximum of $L = 20$ operations per expression, with a number of episodes of 500 to 1000. For the recovery of the Kuramoto system, we use 15 snapshots with time-stamps $t_i = 2i$ for $1 \leq i \leq 15$, and we set $K = 1/3$. The obtained equation, after fitting two regression parameters for 150 samples per screenshot, is given by:

$$\begin{cases} \dot{\theta_1} = 0.0087 + 0.3293 * (\sin(\theta_2 - \theta_1) + \sin(\theta_3 - \theta_1)) \\ \dot{\theta_2} = 0.0087 + 0.3293 * (\sin(\theta_1 - \theta_2) + \sin(\theta_3 - \theta_2)) \\ \dot{\theta_3} = 0.0087 + 0.3293 * (\sin(\theta_2 - \theta_3) + \sin(\theta_1 - \theta_3)) \end{cases} \quad (7)$$

The fitted parameters correspond to $m = 50$ sample points per screenshot.

For the cellular dynamics data, after a pre-processing step using PHATE Moon et al. (2019), the dimension is reduced to $d = 3$ then standard Gaussian noise samples were added to the training set. The following governing ODE system is obtained (after 50 episodes):

$$\begin{cases} \dot{x_1} = \cos(x_2) * x_1 + \cos(x_3) * x_1 \\ \dot{x_2} = \cos(x_1) * x_2 + \cos(x_3) * x_2 \\ \dot{x_3} = \cos(x_2) * x_3 + \cos(x_1) * x_3 \end{cases} \quad (8)$$

Although no coefficient regression has been performed, the model already shows better performance than the competitors as shown in table 2. Given the inductive bias of searching an explicit ODE model, we note that SDFL gives the same model across sample sizes (of the same order, as those illustrated in table 2). For comparison, we used the publicly available implementations of JKOnet and TrajectoryNet, from Bunne et al. (2022) and Tong et al. (2020) respectively. we retrain the models with the architectures and hyper-parameters proposed by the respective authors Tong et al. (2020); Bunne et al. (2022); however, we employ early-stopping to avoid over-fitting to the smaller data-sets[8]. For JKOnet, we use a small regularization parameter $\varepsilon = 0.001$ to make its target closer to the Wasserstein distance. Additionally, to foster reproducibility, a Python implementation of SDFL will be made public following the review of the paper.

## D  Additional numerical experiment details

### D.1  Computational time

We present in tables 3 and 4 a computational time comparison between SDFL, JKOnet and TrajectoryNet. For a fair comparison, all the reported running times are obtained on an Intel(R) Core(TM) i7-7500U CPU. The reported times correspond to training on a sample of $m = 50$ per screen-

| Method | SDFL | JKONet | TrajectoryNet |
|---|---|---|---|
| Time (hours) | 5.8166 | 0.7044 | 5.7709 |

Table 3: Average running time for the Kuramoto system modelling

shot for the Kuramoto system modelling task and $m = 100$ per screen-shot for the scRNA-seq evolution modelling one. It is worth emphasizing that the reported running times for JKONet and TrajectoryNet do not take into account hyper-parameter tuning. And, since SDFL does not require the latter, we believe it is still competitive (for low dimensions).

---

[8]Number of iterations used were 1000 for JKOnet and 1500 for TrajectoryNet

| Method | SDFL | JKONet | TrajectoryNet |
|---|---|---|---|
| Time (hours) | 3.8777 | 0.4696 | 3.8472 |

Table 4: Average running time for the scRNA-seq evolution modelling

### D.2 PERMUTATION-INVARIANCE ABLATION STUDY

For the purpose of illustrating the importance of enforcing permutation-invariance, we report below the performance metric when the latter property is not enforced; for the scRNA-seq evolution modelling task. We denote by `SDFL-WPI` the scheme Without the Permutation Invariance module.

| Method | SDFL | SDFL-WPI |
|---|---|---|
| $\mathcal{W}_2$ metric | 2.0822 | 2.3247 |

Table 5: Prediction loss for the scRNA-seq evolution modelling task

## E DETAILED PSEUDO-CODE

Below, we provide a more detailed pseudo-code of `SDFL` while featuring more clearly the MCTS components.

**Remarks:**

- The back-propagation, such as in line 15, consists in updating the values of the tree states that have been encountered until that step, in that specific episode. Note that the tree states consist of the sequence of operations which have been selected to constitute the chosen expression (until that step).
- The idea is to keep track of the operations (nodes) leading to a good score, and to give them higher chance of being selected in the next rounds.

Symbolic Distribution Flow Learner [extended description]

1: **Inputs:** Number of episodes $N$, number of roll-outs $H$, maximal expression length $M$, elementary functions set $(+, -, \times, \sin, \dots)$, screen-shots $(\hat{\mu}_{t_i,m})_{i,m}$ at $(t_i)_{0 \leq i \leq n}$,

2:

3: **Initialization:**

4: $\rightarrow$ Estimate the value of each operation $(+, -, \times, \sin, \dots)$ as a root node through $H$ stochastic roll-outs;

5: $\rightarrow$ Store these values in $V(0, a)$, where $s = 0$ represents the empty tree state and $a$ the chosen root operation

6: $\rightarrow$ Define $S_{\max} := \max_a V(0, a)$

7:

8: **for** $e = 1, \dots, N$ **do**:

9:

10:     Randomly select a root node and build an expression tree as follows:

11:     **if** Tree is complete **then**

12:         Evaluate the corresponding estimate $\hat{f}$ by computing $S(\hat{f})$;

13:         **if** $S(\hat{f}) > S_{\max}$ **then**

14:             $\rightarrow S_{\max} := S(\hat{f})$;

15:             $\rightarrow$ Back-propagate the obtained value $S(\hat{f})$ by updating the values of $(V(\boldsymbol{s}_p, a))_{p \geq 1}$ where $(\boldsymbol{s}_p)_{p \geq 1} = [a_0, a_1, \dots, a_{p-1}]$ is the finite sequence of encountered tree states before completion;

16:         **end if**

17:     **else**

18:         $\rightarrow$ Run $H$ roll-outs by randomly selecting operations to extend and complete the tree;

19:         $\rightarrow$ Assign a value of $0$ to trees resulting in an inconsistent mathematical expression;

20:         $\rightarrow$ Store the best estimate in $V(\boldsymbol{s}, a_b)$ where $a_b$ is the corresponding best operation;

21:         $\rightarrow$ Back-propagate the obtained value of $V(\boldsymbol{s}, a_b)$ to the encountered tree states $(\boldsymbol{s}_p)_{p \geq 1} = [a_0, a_1, \dots, a_{p-1}]$;

22:         $\rightarrow$ Select the operation $a$ maximizing $UCT(\boldsymbol{s}, a)$ where $\boldsymbol{s}$ is the current state of the tree;

23:     **end if**

24:

25: **end for**

26: **Return:** Most accurate $\hat{f}$ over the $N$ episodes ;

