# OpenReview forum: "WASSERSTEIN-GUIDED SYMBOLIC REGRESSION: MODEL DISCOVERY OF NETWORK DYNAMICS"
_ICLR.cc/2024/Conference — ICLR 2024 Conference Withdrawn Submission_

### Official Review · Reviewer_XDzL · 2023-10-30

**Soundness:** 3 good
**Presentation:** 2 fair
**Contribution:** 3 good
**Rating:** 5
**Confidence:** 3

**Summary:**

The authors developed a white-box Symbolic Distribution Flow Learner (SDFL) based on Monte-Carlo Tree Search (MCTS) with the 2-Wasserstein distance loss for ODE model discovery, given snapshots. Two experiments based on simulated Kuramoto model and reduced single-cell trajectory data have shown the fitting performance of SDFL.

**Strengths:**

SDFL provides a solution to learn ODE models with interpretable functional forms by MCTS-based symbolic regression.

Wasserstein-based loss for MCTS leads to the theoretical analysis of sample complexity of SDFL.

Experimental results have shown that SDFL outperforms JKONet and TrajectoryNet significantly under small and noisy data settings.

**Weaknesses:**

The major concern is that the presentation needs to be improved significantly. For example, the sample complexity proof is mostly based on the reference Shah et al. (2020) while the SDFL takes the UCT-based loss for MCTS, where V(s,a) was not clearly defined in the submission. The provided proof in Appendix B was based on specific loss functional form $h$, which does not seem to be consistent with the text description in Section 3.3. This raises the concern on the validity of the presented theoretical analysis. It is also not really clear how equation (2) is relevant or actually used to derive the presented solution in SDFL. The authors may need to clearly present how MCTS was performed based on Wasserstein distance with stochastic roll-outs.

The presented results on two small systems are based on the fitting performance. It would be interesting to have more comprehensive evaluation, for example, checking the prediction performance in the second experiment?

The authors also need to check on the math notations and reference format. For example, the subscripts in equation (7) in Appendix C need to be fixed and it may be better to use $\theta$ instead of $x$ to be consistent with the notations in Section 5.1.

**Questions:**

1. How coefficient regression was done in the Kuramoto example? In Appendix C, if K is fixed at 1/3, which two regression parameters in the model need to fit?

2. For both experiments, are there any additional constraints imposed in MCTS? How can the functional forms along three dimension be perfectly consistent using MCTS?

3. What is the run-time for SDFL? Is it scalable for high-dimensional ODEs? Is it possible to provide a time complexity analysis?

---

> ### Author Response · Authors · 2023-11-20
>
> We thank the reviewer for the genuine and constructive  comments, which are helpful in improving the paper. Please find our responses below. Revisions have also been made in the paper, in particular the numerical results have been updated.
>
> Weaknesses:
>
> 1. Thanks for this comment. The presentation has indeed been updated and the value of $V(s, a)$ has been explicitly stated. Regarding the proof of proposition 1 (sample complexity), in fact MCTS is a maximization algorithm, so minimizing the loss $L$ which is the initial goal, is equivalent to maximizing $\frac{1}{1+ L}$. Hence, the latter expression was used for the algorithm and that is why it is the one  used for the proof. This information has now been explicitly stated in subsection 4.2. Regarding equation $(2)$, it was provided (in the background material) to give more context on the problem, but it is -indeed- not explicitly  needed for the results.
>
> 2. Thanks for this comment. The numerical results are actually not fitting performance but rather prediction performance, on an unseen test data-set (as mentioned in the captions of the tables).
>
> 3. Thank you for your attention to these details. We appreciate it. We have updated the notations accordingly.
>
> Questions:
>
> 1. Thanks for this question. One coefficient was used for $K$ and one for the intercept (i.e. the additive constant known as natural frequency).
>
> 2.  Thanks for these remark and questions. That was enforced using the permutation invariance trick, since such a structure is what's expected for a network model. We added in appendix D.2 a model obtained without enforcing permutation invariance, and the performance is lower. Additionally, we note that the task of modelling scRNAseq data has been previously shown to benefit from network modelling in static settings, such as in [1] and [2].
>
> 3. Many thanks for these relevant points. A comparison of computational time between the different methods has been added in appendix D.1. Scalability of symbolic search to high dimensions is still an open problem (in general) as stated in the updated conclusion and left for future work.
>
> References
>
> [1] Wang, Juexin, et al. "scGNN is a novel graph neural network framework for single-cell RNA-Seq analyses." Nature communications 12.1 (2021): 1882.
>
> [2] Van Dijk, David, et al. "Recovering gene interactions from single-cell data using data diffusion." Cell 174.3 (2018): 716-729.

---

### Official Review · Reviewer_97PM · 2023-10-30

**Soundness:** 3 good
**Presentation:** 3 good
**Contribution:** 3 good
**Rating:** 5
**Confidence:** 3

**Summary:**

Authors propose the combination of Neural ODE with symbolic regression using MCTS and Wasserstein loss to learn dynamical systems under an observational setting, where longitudinal data is measured at random times. The problem of learning the dynamics from the observed snapshots is a type of inverse problem which is in general non-identifiable.

The method requires a set of symbolic operators to be considered for building the symbolic expressions. The method uses MCTS and the set of provided operators to search for an analytical formula of the ODE system, and the Wasserstein loss is used to measure the discrepancy between the predictions and observations.

The method was tested on a toy dynamical model and on single cell data.

**Strengths:**

The major strength of this approach is the combination of Neural ODEs and the MCTS component to find symbolic expressions for an ODE system, in which observations of the system come at random times. Being able to learn more interpretable models in this setting is the major strength.

**Weaknesses:**

The primary shortcoming of this paper lies in its evaluation. It seems that the primary bottleneck of the method is its scalability since learning symbolic expressions is a complex problem. As such, the method is only suitable for very low-dimensional data. In this respect, the authors only provide a single toy model (3 dims and 15 observations) where the equations are known, and a basic comparison against JKOnet is offered. In this setting, it is expected that the method performs better, as the operators given to the method are part of the equations. While this analysis is adequate as a sanity check, it doesn’t offer much information about the overall capabilities of the method and it's hard to judge the method's effectiveness. Moreover, running times are not mentioned, so it's unclear how the costs of running SDFL compare to JKOnet or TrajectoryNet. Only the Wasserstein error is provided as a metric for comparison.

The authors also present a second experiment on single-cell dynamics. However, I'm skeptical about the usefulness of this method in this context, given the vast number of dimensions typical of such problems. Here, 20K measurements of genes are available for hundreds or thousands of cells at different time points. This doesn't seem like an ideal setting for this method. To handle the complexity of this dataset, the authors analyzed the data in a very low-dimensional space (d=3), mentioned only in Appendix C, with little detail on how this was accomplished. For instance, with JKOnet, the authors pick the 4K most variable genes and then use the first 20 principal components of PCA on those genes. I assume the authors used a similar approach to reduce from 20K dimensions to just 3. However, even if the authors argue that one of the method's strengths is interpretability, it's hard to see how one can interpret the equations of a 3D embedding of a dynamical system. Unless the authors can offer some interpretation or highlight which genes are the main drivers of changes across timepoints in ways that other methods can't, I question the utility of this example. A more in-depth analysis would be beneficial, illustrating, on small-scale problems, how this method might yield better or more interpretable results than its peers.

**Questions:**

The proposed method is interesting and appears to offer some advantages over two other SOTA methods in a very ad-hoc setting. Although the method holds promise for practical application, the authors fall short of demonstrating this in a broader experimental context using realistic yet low-dimensional datasets. Since interpretability ("white-box") is one of the major claims, I believe situations where authors must artificially reduce the problem's dimensionality should be sidestepped unless this process improves interpretability. Can you explain in more detail how one can interpret the equations of a 3D embedding of a dynamical system?

The authors should illustrate the method's performance on low-dimensional datasets across various settings, especially when partial knowledge of the symbolic operators is provided to the algorithm. Emphasis should be placed on interpreting the learned dynamics rather than just showcasing a decrease in Wasserstein loss. Additionally, a thorough description of the competitor methods' configurations is expected to be provided. How comprehensive is the comparison? Did the authors optimize the hyperparameters for the competing methods? this information is not available.

Can the method learn a good approximation of the dynamics if a cosine is provided instead of a sine as part of the input? How would it perform with only partial prior knowledge compared to JKOnet, which doesn’t need this prior knowledge?

The manuscript lacks a comprehensive comparison in terms of scalability, making it challenging to assess the method's practicality. It seems to be challenging to apply the method even in a low dimensional setting, since the authors resort to a two-step procedure: the initial step identifies the structure, and the subsequent step estimates coefficients. Does this suggest potential scalability issues even with modest sample sizes?

As acknowledged by the authors, MCTS was used before in the context of symbolic regression, as well as Neural ODEs + Symbolic regression. Some additional refs that can be interesting for the authors are https://arxiv.org/abs/1901.07714 and https://arxiv.org/abs/2202.02435.

---

> ### Author Response · Authors · 2023-11-20
>
> We thank the reviewer for the constructive comments, which are helpful in improving the paper. Please find our responses below. Revisions have also been made in the paper, in particular the numerical results have been updated.
>
> Weaknesses:
>
> 1. Thank you for this comment. As mentioned in appendix C, the operators that were given to the search algorithm are
>  $$\{+, - , \times, \frac{ \cdot  }{ \cdot }, \cos, \sin, \exp \}.$$ This leads to a large search space of the order of millions of possible expressions, for a predefined maximum length of 10 to 20 symbols. Yet, the proposed method is able to recover the model with very sparse observations and no use of the velocities. Symbolic search usually requires a much higher sampling frequency, and does not handle randomness (see for e.g. \url{https://par.nsf.gov/servlets/purl/10357448}). We believe our method is the first that is able to address the important challenges of data-sparsity and randomness. Not only that, but the method also shows promising improvement in performance in comparison to the state-of-the-art in trajectory inference. Computational time comparison has also been added in appendix D.1. As mentioned by the reviewer, scaling symbolic learning to high dimensions is still an open problem, which is out-of-the-scope of this paper. Regarding the evaluation metric, it measures the ability of the model to accurately predict the system under study. It is also the metric that was used by previous works (e.g. JKOnet and TrajectoryNet). Additionally, as kindly acknowledged below by the reviewer, we also propose a real-world experiment where the equations are not known, which we discuss below.
>
> 2. Many thanks for this comment. More details on the pre-processing of the data have been added in the updated version, which we briefly mention here. (In the previous version, the embedding technique was only mentioned in Appendix C). The dimension was reduced using the recent dimensionality reduction technique PHATE, proposed in [4]. PHATE is specifically designed to preserve maximum variablity in the low-dimensional space, while allowing for intuitive visualization, and therefore interpretation. Given that visualization is more intuitive in dimension 3, we believe learning a model on top of that would lead to improved interpretation. We also note that TrajectoryNet paper uses a dimension $d=5$, which is of the same order of what we use. Furthermore, We emphasize and kindly remind the reviewer that the pre-processing is not part of our method, nor was it part of other trajectory inference algorithms such as JKOnet and TrajectoryNet. Hence, the " white-box " claim mostly concerns the main task addressed in the paper, i.e. devising a model that is able to predict probability distribution flows accurately.

---

> > ### Author Response · Authors · 2023-11-20
> >
> > Questions:
> >
> > 1.    Thank you for this positive comment and question. The scRNAseq dataset was chosen because it is a standard benchmark on the topic of trajectory inference, while still illustrating the potential of the proposed method for low-dimensional problems - although it entails a pre-processing step. Moreover, reducing the dimension of a system allows for intuitive visualization as mentioned above, leading to easier interpretation.
> >
> >   2.  Thank for this comment. Additional information on the training of the competitor methods has been added in the updated manuscript. The comparison is based on based on retrained models with the architectures and hyper-parameters proposed by the respective authors [2, 3]; however, we employ early-stopping to avoid over-fitting to the smaller data-sets. In the case where partial knowledge of the operations is available, it is rather obvious that the performance of SDFL could only improve, which is why we believe it is more challenging to test it in a setting where such information is not available (as we did). Comparison of the Wasserstein distance illustrates the accuracy of the method, which is the first criterion usually looked at. Moreover, it is the way different methods have been compared and evaluated in trajectory inference. Interpretation of the obtained dynamics is left for domain experts, who are more familiar with the specific application. But, inspired by the reviewer's comment, we restate in the updated manuscipt that a white-box model learner (like the one we propose) allows for the study of qualitative properties such as synchronization of the learned models, also enabling (improved) interpretation.
> >
> > 3. Thank you for this question. As mentioned above, the operations provided to the search algorithm were
> >  $$\{+, - , \times, \frac{ \cdot  }{ \cdot }, \cos, \sin, \exp \}.$$ Hence, it did not have any special prior knowledge that sine should be used, as opposed to cosine for instance.
> >
> > 4. Thank you for this remark and question. A computational time comparison has been added in appendix D.1. However, the point of the method is to be used in settings where data is scarce and costly to obtain, not really with large sample sizes.
> >
> > 5. We thank the reviewer for these relevant references, which we have cited in the updated version of the manuscript.
> >
> >
> > References
> >
> > [1] Qian, Zhaozhi, Krzysztof Kacprzyk, and Mihaela van der Schaar. "D-code: Discovering closed-form odes from observed trajectories." International Conference on Learning Representations. 2022.
> >
> > [2] Tong, Alexander, et al. "Trajectorynet: A dynamic optimal transport network for modeling cellular dynamics." International conference on machine learning. PMLR, 2020.
> >
> > [3] Bunne, Charlotte, et al. "Proximal optimal transport modeling of population dynamics." International Conference on Artificial Intelligence and Statistics. PMLR, 2022.
> >
> > [4] Moon, Kevin R., et al. "Visualizing structure and transitions in high-dimensional biological data." Nature biotechnology 37.12 (2019): 1482-1492.

---

> ### Comment · Reviewer_97PM · 2023-11-22
>
> I would like to thank the authors for the taken time and effort to address some of the questions. While they offer a bit more insight, they do not sufficiently address the raised issues to the extent that it warrants change of rating. Some of the answers address the questions only partially, for example not addressing direct requests for comparison. The authors also don't address more explicitly the pointed domain specific interpretability issues, a major upside of symbolic methods, beyond simple single metric comparison. Furthermore, a standard benchmark dataset hasn't been established nor routinely used in the single cell community and there are large number of datasets and findings obtained from trajectory inference methods that the outcome of the approach can be compared to on the basis of various metrics capturing different aspects of the results. See for example https://doi.org/10.1038/s41587-019-0071-9.

---

> > ### Author Response · Authors · 2023-11-22
> >
> > We thank the reviewer for their time and feedback. As for the remaining concerns:
> >
> > - We stated that the real-wold dataset/task, that we used, is a standard benchmark in the sense that
> > most recent machine learning works on trajectory inference (on distributions) use it as the real-world task to test
> > their methods, e.g. [1, 2, 3, 4, 5]
> >
> > - Regarding the comparison in case where partial knowledge of the operators is known, we thought we addressed it
> > but we can clarify more. When some operations are known, the elementary starting set is reduced
> > hence the search space is reduced, therefore necessary making the task less challenging.
> >
> > - Regarding the interpretability, we kindly ask the reviewer to note that each single aspect of interpretability
> > (for instance synchronization) is worth a full study for its own sake such as in [6, 7, 8], which is why we think it is
> > out-of-the-scope of this paper.
> >
> > We really appreciate the time you have taken for this discussion.
> >
> > Best regards,
> > The authors
> >
> > [1] Chizat, Lénaïc, et al. "Trajectory inference via mean-field langevin in path space." Advances in Neural Information Processing Systems 35 (2022).
> >
> > [2] Hashimoto, Tatsunori, David Gifford, and Tommi Jaakkola. "Learning population-level diffusions with generative RNNs." International Conference on Machine Learning. PMLR, 2016.
> >
> > [3] Tong, Alexander, et al. "Trajectorynet: A dynamic optimal transport network for modeling cellular dynamics." International conference on machine learning. PMLR, 2020.
> >
> > [4] Bunne, Charlotte, et al. "Proximal optimal transport modeling of population dynamics." International Conference on Artificial Intelligence and Statistics. PMLR, 2022.
> >
> > [5] Huguet, Guillaume, et al. "Manifold interpolating optimal-transport flows for trajectory inference." Advances in Neural Information Processing Systems 35 (2022).
> >
> > [6] Ódor, Géza, and Jeffrey Kelling. "Critical synchronization dynamics of the Kuramoto model on connectome and small world graphs." Scientific reports 9.1 (2019).
> >
> > [7] Menara, Tommaso, et al. "Functional control of oscillator networks." Nature communications 13.1 (2022).
> >
> > [8] Ly, Cheng, and Seth H. Weinberg. "Analysis of heterogeneous cardiac pacemaker tissue models and traveling wave dynamics." Journal of Theoretical Biology 459 (2018).

---

### Official Review · Reviewer_QgHg · 2023-10-30

**Soundness:** 3 good
**Presentation:** 2 fair
**Contribution:** 2 fair
**Rating:** 5
**Confidence:** 3

**Summary:**

The paper studies the symbolic learning of probability flows –the flows are originated by a distribution of initial conditions that the ODE has. The solution is based on a Monte Carlo Search Tree solution to solve a given optimization problem, and theoretical guarantees is given for the approximation of the optimization problem given that we only use samples of the trajectories, and for sample complexity in finding a close to optimal solution. Experiments in both different settings are given to show its competitive performance -- one where the "ground-truth
 ODE is known and one where isn't.

**Strengths:**

-> An important strength is combining MCTS methods for symbolic regression to the minimization of a Wasserstein-based metric for the probability flows of the system. From what I see in the paper, this is the first time such formulation has been done.

-> It is great that the paper has both theory and experiments, since both are needed due to the nature of the problem being resolved.

-> Experimental results show competitive performance in discovering the underlying dynamics of a Kuramoto oscillator network and of single-cell population dynamics. The method is compared to recent methods in the literature.

I believe the paper is of interest to the community.

**Weaknesses:**

The paper has considerable room for improvement in the presentation of its results, both theoretical and experimental, as well as in literature review, and the writing in general. I am expanding on my concerns below.

-> In subsection 3.1 I would imagine that assumptions on regularity or continuity on the vector field $f$ must be enforced, to avoid, for example, solutions that escape in finite-time (will diverge at some finite time before $T$), e.g., $\dot{x}=x^2$.

-> Is $s$ in Theorem 1 any real number that we must choose to be greater than $d$, or is it some parameter coming from the setting from the sections before Theorem 1?

-> Proposition 1 requires more explanation. First of all, would it be possible to include a mathematical expression that defines what an $\epsilon$-optimal solution is? Would it be possible to reference the objective that is being optimized? Moreover, two expressions are introduced whose definition is nowhere to be found: “maximum allowed expression length” and “size of the chosen elementary function set”.

-> Regarding the main algorithm (subsection 4.2): It is mentioned that a “permutation invariant
aggregation operation (e.g. sum)” is included to reduce the search space to the MCTS. So, how does such invariants help reducing such search-space? I guess such operation is part of a leaf of the MCTS, right? Should it be highlighted in Figure 1 or in Algorithm 1?

-> Something which makes not much sense to me is the fact that the introduction section makes it clear that there is an interest in analyzing network systems and much of the paper’s inspiration comes from it. However, in the next sections until the experimental section, there is not mentioning of network systems playing a particular role in the derivations of the theoretical results or even the algorithm – perhaps with the exception of the brief mentioning of “permutation invariant aggregation operation”, which, as I mentioned before, is barely explained and noticeable in the algorithm and the paper itself. Is there any explanation missing regarding the role of having network systems in the technical derivations and algorithm? I don’t really see the connection –as far as how things look to me, all the derivations in the paper and algorithm should work for appropriate general vector fields $f$ that are not necessarily corresponding to network systems of some sort.

-> The description of the Monte-Carlo tree search for symbolic regression is not self-contained in my opinion and requires more explanation. Would it be possible to describe it as an algorithm, with “for loops” and precise instructions? I need to find answer to questions such as: Are the nodes, especially the first node of the tree, considering both binary and unitary operations? What is the “predefined measure of accuracy” on step 2 (Simulation)? Is this “back-propagation” step similar to the one in dynamic programming? How many times is step 4 (Selection) repeated?
I believe that an example with a graphic will greatly help.

-> Why is TrajectoryNet not depicted in Table 1? Its comparison must be added to understand fully the comparison of SDFL.

-> Authors must include a detailed comparison between their solution and “TrajectoryNet” and “JKOnet”, their similarities and difference in their algorithms. This is crucial and currently absent from the paper. For the Kuramoto experiment I would expect TrajectoryNet to perform better since it seems to be based on neural networks, which perform well when there is a lot of data.

-> Why does SDFL have better performance in the simulations (considering that a comparison with TrajectoryNet is absent in the first experiment, as pointed above)? An explanation or even a speculation should be given, since it highlights how different the method done by the authors are with respect to the others.
Also, do the other methods also incorporate some permutation invariant properties in their methods?
Also, is there perhaps a metric (besides the Wasserstein difference) under which the other methods perform better? What do they use in their own papers to assess accuracy?

-> What happens when you remove the aggregation operation (which is permutation invariant) in both experiments? I want to see how much it really plays a role; this is a very important ablation study whose results must be in the paper. Moreover, the last experiment had nothing to do with networks, so I wonder what role adding such additive operations was important.

-> So, for the Kuramoto oscillator experiment, what is the “discovered” ODE according to the paper’s method SDFL? The whole point of the paper is being able to show that the method discovers the symbolic expression of the dynamical system, and this is absent in the paper! It would be nice to see what it says on the other experiment too.

-> The simulations don’t show anything regarding the computational time taken by all the methods. Please, include this. Maybe there is a trade-off between accuracy and performance in your algorithm.

==

-> In the introduction, properties such as “permutation invariance” and “partial observability” are mentioned without any reference to what they mean. Their meaning should be stated, at least from a qualitative perspective, as well as why they are important or show up in the problem being studied of network flows.

**Questions:**

Please, see the "Weaknesses" section above.

---

> ### Author Response · Authors · 2023-11-20
>
> We thank the reviewer for the constructive and detailed comments, which are very helpful in improving the paper. Please find our responses below. Revisions have also been made in the paper, in particular the numerical results have been updated.
>
> 1. Thanks for this very good remark. The vector fields leading to solutions which explode before the time horizon $T > 0$, are discarded in the assumption that the solution is defined on the compact interval $[0, T]$. From a practical perspective, since the system is observed at predefined observation times, if an expression such as $\dot{x} = x^2$ is wrongly selected -for evaluation- then the loss function between the observed values and the predicted ones will be large as the time step approaches the explosion time and such an expression will be discarded. On the other hand, if it is the right equation then the observation times would be such that the explosion time does not belong to the interval $[0, T]$. Additionally, the search space is reduced (by construction in the method) to functions $f$ which have an explicit analytical expression and therefore smooth (i.e. infinitely differentiable) on their domains.
>
> 2. Thanks for this comment. We have updated Theorem 1 using results more specific to $\mathbb{R}^d$, and the variable $s$ is no longer needed.
>
> 3. Thanks for this comment. A definition of $\varepsilon-$optimal solution has been added (as a footnote to the proposition, given that it's usually considered a common notion) and the function to be maximized has clearly been restated in the paragraph preceding the proposition (i.e. subsection 4.2). Regarding the expressions “maximum allowed expression length” and “size of the chosen elementary function set”, they refer to the previously explained fact that the search algorithm is based on a set of elementary operations/functions $(+, \times, \cos, \dots)$ with the goal of combining them to output an accurate model. This entails fixing the maximal length that a mathematical expression is allowed to have (which is denoted $M$ in the updated version, in subsection 3.3 stochastic roll-outs paragraph), to define a relevant search space.
>
> 4. Thanks for this comment. An aggregation operation (ensuring permutation invariance) reduces the search space in the sense that, since the expected expression will be symmetrical (or more precisely invariant) with respect to permutations of variables, the search can be reduced to looking for the part of an expression that depends only on two variables, but generates the full expression. For example, suppose the (first component of the) optimal expression is $\dot{x_1} = x_1 \cdot x_2 + x_2 \cdot x_3 + x_1 \cdot x_3$, then the algorithm can look for $x_1 \cdot x_2$ and deduce the rest. This has been added to subsection 4.2.
>
> 5. This is a very good point. Indeed, the theoretical results apply to any vector field (satisfying the stated assumptions). On the other hand, in the algorithm, depending on whether the system is a network or not, one may use the permutation invariance trick. Regarding the motivation mentioned in the introduction, it comes from the fact that network systems are an important use-case of the proposed method, which is why the numerical evaluation is dedicated to such systems.
>
> 6. Thanks for this comment. The description of MCTS in subsection 3.3 has been clarified and a more detailed pseudo-code of SDFL -clearly featuring how MCTS components are used- has been added in appendix E. As for the specific questions the reviewer is asking:
> a) Indeed, all the nodes consider both binary and unitary operations, except leaves.\\
> b) The predefined measure of accuracy is $S(\hat{f})$  (as defined in subsection 4.2). \\
> c) The "back-propagation" operation consists in updating the encountered tree state values (i.e. node values) -denoted $V(s, a)$ where $a$ is the  operation selected at tree state $s$- for relevant selection in the next rounds. Hence, it is rather different from the one in dynamic programming.
> d) Selection (step 4 in the first manuscript) is repeated until the tree is complete and corresponds to a coherent mathematical expression, or until the length of the latter reaches the predefined maximal length (denoted by $M$, see subsection 3.3), for as many episodes as chosen in the input of the global algorithm.
>
> 7. Thanks for this comment. TrajectoryNet has also been added in the first experiment (as well) in the updated manuscript.
>
> 8. Thank you for this comment. Such a comparison has been added in section 5. Both the other methods (which are the current state-of-the art) are based on neural networks. Still, they do not perform as well as SDFL in either of the experiments because the datasize is not large enough. In the corresponding papers which introduced JKOnet and TrajectoryNet, the authors use data-sizes of the order of 2000-3000 per screen-shot, whereas we use at most 300 samples per screenshot.

---

> > ### Author Response · Authors · 2023-11-20
> >
> > 9. Thanks for these questions. As already mentioned in subsection 5.2, please note that, neural net based approaches require sufficiently large amounts of data to reach their optimal performance. The other methods do not allow for the incorporation of permutation invariance, which is one their shortcomings. Regarding evaluation metric, as stated in section 5, the other methods also use the Wasserstein metric for their evaluation. Unlike TrajectoryNet, JKOnet uses a regularized version of the metric, known as the Sinkhorn divergence, but such a measure is not a distance, and it's only an approximation of the Wasserstein metric. Please note that the other popular measure of disparity between probability distributions, namely the Kullback–Leibler divergence is not defined when the distributions do not have the same support, as in our case.
> >
> > 10. Thanks for this comment. In case the aggregation operation is removed, a much larger search space is considered and the algorithm requires much more computational time/ resources to recover the Kuramoto model. It might fail to recover it, if the time horizon and the number of search episodes is not large enough. For the second experiment, a more precise description of the data has been added. Indeed, the method is actually applied to body single-cell RNA sequencing (scRNA-seq) data (following TrajectoryNet and JKOnet papers). For such a system, network modelling is actually most relevant as demonstrated by domain-specific publications such as [1, 2]. Similarly, when the aggregation module is removed for the second experiment, the model that is obtained does not perform as well as with the aggregation. It is added for illustration in Appendix D.2.
> >
> > 11.  This is a good point. However, given the space constraints, the discovered models had been added in Appendix C. We believe it is still (unfortunately) not possible to add them in the main text, given the space they would take. However, a clear reference to them has been added in the first paragraph of section 5.
> >
> > 12. Thanks for this remark. We added a comparison between computational time of SDFL and the competitor approaches in appendix D.1. Regarding the mentioned trade-off, there is indeed one between performance and computational resources/time. Below a certain time horizon for the symbolic search, the method does not necessarily find the most accurate model. We emphasize though that such trade-offs are present in any approach. For example, a neural network which is only trained for a small amount of time will perform much worse than one trained for an optimal amount of time. Additionally, we believe the method we propose is still relevant regardless of that trade-off, since its main strengths are robustness and sample-efficiency (i.e. its ability to learn from a much smaller amount of data).
> >
> > 13.  Thanks for this comment. Permutation invariance is the fact that the state of a node depends on those of its neighbors in the same way whatever is the chosen order of the neighbors. This is a well known property of network systems, which should be enforced in order for the model to be suitable and potentially to reduce the compuational cost. Please, see [3] for reference. Regarding partial observability, it commonly refers to a problem which has an uncertainty component, but also one where observations are sparse and scattered, as is the case in the setting we address (the observation of a few screenshots across time in contrast to observing a system at a high sampling frequency).
> >
> > References
> >
> > [1] Wang, Juexin, et al. "scGNN is a novel graph neural network framework for single-cell RNA-Seq analyses." Nature communications 12.1 (2021): 1882.
> >
> > [2] Van Dijk, David, et al. "Recovering gene interactions from single-cell data using data diffusion." Cell 174.3 (2018): 716-729.
> >
> > [3] Bronstein, Michael M., et al. "Geometric deep learning: Grids, groups, graphs, geodesics, and gauges." arXiv preprint arXiv:2104.13478 (2021).

---

### Official Review · Reviewer_h3SD · 2023-10-31

**Soundness:** 2 fair
**Presentation:** 2 fair
**Contribution:** 2 fair
**Rating:** 3
**Confidence:** 3

**Summary:**

The article presents a methodology for symbolic regression of ODE-driven probability distributions. The method is white box and the relies on a Wasserstein loss

**Strengths:**

Symbolic learning is of interest across the ML community and data science practitioners due to transparency requirements. Furthermore, combining this with a Wasserstein loss is a challenge and, thus a strength of this paper.

**Weaknesses:**

Although the article's aim is of great interest, the proposal is not explored in depth for this venue. For instance:

- One motivation for studying symbolic regression (in the abstract) is interpretability. However, no interpretation is provided in the experimental/validation part. Furthermore, though the proposal is referred to as a _white box_ it is never inspected.

- There are some referencing issues: Sec 4 refers to a Fig 4, but there are only three figures in the paper. As mentioned Fig 4.2 does not exist.

- The article dedicates a fair share of its extension to the background: up until page 6 is previous work. Though this can be useful for those unfamiliar with the required background, it leaves little space to discuss the proposal in more detail. For instance, the core contribution of the article (referred to as Technical Approach in Sec 4) is only contained in 1.5 pages.

- Following the above point, one would expect that this is an experimental contribution; however, the experimental validation is rather limited. Both the synthetic and real-world examples are implemented for different amounts of datapoints; however, the reason for a varying-size dataset is not justified, and in particular, it is not stated whether the performance is expected to increase or decrease with more data. Also, there are no error bars.

- Fig 3, which shows the experiments of the real-world dataset, provides no insight into the contribution. In particular, 3b only compares the Euclidean and Wasserstein distances in two cases.

- There is no reference or discussion about the computational cost of the presented strategy

**Questions:**

Please refer to the comments in the previous section

---

> ### Author Response · Authors · 2023-11-20
>
> We thank the reviewer for the constructive comments, which are helpful in improving the paper. Please find our responses below. Revisions have also been made in the paper, in particular the numerical results have been updated.
>
> 1. One motivation for studying symbolic regression (in the abstract) is interpretability. However, no interpretation is provided in the experimental/validation part. Furthermore, though the proposal is referred to as a white box it is never inspected.
>
> Thanks for this comment. The proposed scheme yields white-box models, in the sense that the models are explicit ODEs (reported in appendix C), which can be interpreted by domain experts. This contrasts with the current state-of-the-art competitors, which are based on neural networks, and which neither allow interpretation nor qualitative studies such as synchronization, stability, etc., although the latter are also crucial in practice. An explicit reference to the equations obtained by SDFL, reported in appendix C, has been added in the first paragraph of section 5.
>
> 2. There are some referencing issues: Sec 4 refers to a Fig 4, but there are only three figures in the paper. As mentioned Fig 4.2 does not exist.
>
> Many thanks for your attention to that. We have corrected these issues and we will do another round of proofreading to make sure any typos are fully eliminated.
>
> 3. The article dedicates a fair share of its extension to the background: up until page 6 is previous work. Though this can be useful for those unfamiliar with the required background, it leaves little space to discuss the proposal in more detail. For instance, the core contribution of the article (referred to as Technical Approach in Sec 4) is only contained in 1.5 pages.
>
> This is a good point. We have reduced the related works section accordingly. As for the general setup and Wasserstein guidance sub-sections, we kindly ask the reviewer to note that they introduce both the required technical setup and the proposed loss function (which is an important and novel component of the approach).
>
> 4. Following the above point, one would expect that this is an experimental contribution; however, the experimental validation is rather limited. Both the synthetic and real-world examples are implemented for different amounts of data-points; however, the reason for a varying-size dataset is not justified, and in particular, it is not stated whether the performance is expected to increase or decrease with more data. Also, there are no error bars.
>
> Thanks for these comments. Please note that before the numerical results, the contribution includes two theoretical results, whose proofs are postponed to appendices A and B. Regarding the sample-size, the reason why it was varied is because the method tackles the small data regime, with applications where data measurements are highly costly. This remark has been added in the updated version along with comments on the evolution of performance with the increasing sample size. Additionally, error bars have been added.
>
> 5. Fig 3, which shows the experiments of the real-world dataset, provides no insight into the contribution. In particular, 3b only compares the Euclidean and Wasserstein distances in two cases.
>
> Thanks for this comment. In fact, figure 3(a) illustrates the closeness of the estimated curve with the ground truth one for the Kuramoto model task. On the other hand, figure 3(b) illustrates the robustness of the Wasserstein distance across time, as it is a key ingredient of the proposed loss function $\hat{L}_{m, n}$. Both of these facts were stated in subsection 5.1. Additionally, the equations obtained by SDFL were reported in appendix C.
>
> 6. There is no reference or discussion about the computational cost of the presented strategy.
>
> Thanks for this comment. We have added a comparison of the computational time of the method with the competitors in appendix D.1.